# Ultrasound-Guided Hook-Wire Localization for Surgical Excision of Non-Palpable Superficial Inguinal Lymph Nodes in Dogs: A Pilot Study

**DOI:** 10.3390/ani10122314

**Published:** 2020-12-07

**Authors:** Alessio Pierini, Veronica Marchetti, Matteo Rossanese, Riccardo Finotello, Andrea Cattai, Guido Pisani

**Affiliations:** 1Department of Veterinary Sciences, University of Pisa, 56122 Pisa, Italy; veronica.marchetti@unipi.it; 2Centro Veterinario Dott. Pisani-Carli-Chiodo, 19034 Luni, Italy; guido.pisani@gmail.com; 3Department of Small Animal Clinical Science, Institute of Infection, Veterinary and Ecological Science, University of Liverpool, Neston CH64 7TE, UK; matteo.rossanese@gmail.com (M.R.); r.finotello@liverpool.ac.uk (R.F.); 4Clinica Veterinaria Concordia, 30026 Portogruaro, Italy; andrea.cattai.vet@gmail.com

**Keywords:** imaging-guide localization, hook-wire, lymphadenectomy, lymph node tracking, nodal metastasis, staging, superficial inguinal lymph node

## Abstract

**Simple Summary:**

The evaluation of regional/sentinel lymph nodes (LN) plays an important prognostic role and assists the clinical decision making in canine cancer patients; thus, surgeons are frequently asked to perform regional/sentinel lymphadenectomies together with management of the primary tumor sites, even when these are non-palpable and/or normal size LN. Several techniques have been described to localized regional/sentinel LN in a pre-surgical setting. The wire-guided localization is commonly used to localize non-palpable breast masses in women, with the aim to reduce surgical dissection and excision of unnecessary tissue and to reduce surgical time. This study describes the use of an ultrasound-guided hook-wire localization method to facilitate intraoperative localization of non-palpable superficial inguinal LN. The results of the present study suggest that this method is safe and effective and may allow a high rate of successful superficial inguinal LN excisions in dogs.

**Abstract:**

The evaluation of loco-regional lymph nodes (LN) plays an important prognostic role and assists the clinical decision making in canine cancer patients. Excision of non-palpable LN can be challenging. The aim of the study was to evaluate surgical time, successful excision rate and surgical complications associated with the use of an ultrasound-guided hook-wire (UGHW) LN localization method for non-palpable superficial inguinal LN (SILN) in dogs. Dogs that presented for excision of non-palpable SILN, performed with the aid of an UGHW placement, were enrolled. Information including signalment, SILN width, UGHW placement and surgical procedure time, hook-wire position, successful excision and intra- and post-operative complications were reviewed. Seventeen dogs were enrolled. Median LN width was 3 mm (range 2–11). Median time of preoperative UGHW placement and surgical LN excision was 8 min and 15 min, respectively. Successful SILN excision was achieved in all cases. Two minor intra-operative (hook migration and wire fragmentation) and one minor post-operative complications (seroma) were observed. No major intraoperative or post-operative complications occurred. The UGHW LN localization method is safe and effective and may allow a high rate of successful SILN excisions in dogs. This method has the potential to facilitate LN excision for other superficial LN locations.

## 1. Introduction

The status of loco-regional lymph nodes (LN) in dogs with malignancies plays a pivotal role in clinical decision making [1,2], and the treatment of metastatic LN is associated with longer survival times in dogs diagnosed with mast cell tumor and anal sac apocrine gland adenocarcinoma [3,4,5].

Nowadays, surgeons are more frequently asked to remove one or more non-palpable LN to better characterize malignancies such as head cancers, cutaneous mast cell tumor and oral melanoma [1,2,6]. Bilateral mandibular and medial retropharyngeal lymphadenectomy have been suggested to provide more accurate information in dogs with head cancers, being able to unmask metastatic disease in normal sized LN with minor post-operative complications [1,7]; however, little is known on morbidity and success of lymphadenectomy of non-palpable peripheral LN at other sites.

In dogs, superficial inguinal lymph nodes (SILN) are located deep to the inguinal mammary gland in females and at approximatively the level of the bulbus glandis in males [8]. A single SILN is more often present, however two or more SILN may be present simultaneously [8,9]. Even if superficial and considered palpable by some authors [10,11], normal-size SILNs, in the authors’ experience, are unlikely to be distinguishable from subcutaneous or fat tissue in dogs, especially if they are overweight. Moreover, nodal localization can be time consuming, requiring extensive tissue dissection.

In a previous study SILN have been easily identified with sonography in forty-seven (96%) of 50 dogs, highlighting the potential role of ultrasound in the pre-operative localization of SILNs [9].

Wire-guided localization is a method commonly used to locate non-palpable breast masses in women [12]. The wire is most commonly applied by ultrasound or mammographic guidance and is able to guide the surgical dissection to the lesion reducing associated morbidity and shortening the surgical time [13,14,15]. In human medicine, this technique has also been reported to facilitate excision of small pulmonary nodules and non-palpable soft tissue sarcoma [13,16]. Recently, a small case series described the use of ultrasound-guided hook-wires (UGHW) to locate regional LN in four dogs and results suggested that this method was safe and effective [17].

The aim of the study was to describe the clinical application of a UGHW LN localization method for the excision of non-palpable SILN in dogs and to report the successful excision rate and surgical complications associated with this technique.

## 2. Materials and Methods

Dogs affected by different tumor histotypes presented at two referral hospitals for diagnostic or therapeutic lymphadenectomy of superficial inguinal lymph node with the aid of UGHW placement between August 2019 and May 2020 were enrolled. This study was approved by the University of Pisa animal welfare and research ethics committees (48/2019) and the University of Liverpool Veterinary School Research Ethics Committee (VREC932). Dogs were eligible for inclusion if the SILN to be excised was deemed non-palpable. Every LN was excised via a dedicated surgical approach and lymphadenectomy was performed by five different surgeons (two board-certified, one resident and two experienced surgeons). Information retrieved from the records included breed, age, sex and sexual status, body condition score (body condition score (BCS), range 1–9), body weight, primary tumor histotype, LN width, time of UGHW placement, surgical procedure, lymphadenectomy time, total surgical and anesthetic time, hook-wire position, results of histopathological examination of the excised LN, and intra- and post-operative complications. The position of the hook was considered perinodal, if the hook tip was less than 5 mm far from the node, or intranodal if the tip of the hook was within the LN. When the hook dislodged or was found far from the node, or was not to be found it was considered out of the node. Intra- and post-operative complications were classified as major if they required further surgical or medical treatment to resolve (i.e., severe hemorrhage that needs surgical treatment or blood transfusion, surgical site infection, dehiscence of the surgical site), and as minor if they required no treatment to resolve. Minor intra-operative complications included also hook dislodgement and hook-wire fragmentation. Dogs were clinically re-checked seven and 14 days after surgery and dogs with a minimum follow up of less than fourteen days were excluded. All dogs underwent a physical examination and blood test analysis before the procedure was performed and written informed consent was taken beforehand from all the owners.

For all cases, a 20 G × 7 cm single or double hook non-repositionable needle for mammary nodules localization (IM-IMX, OIM2007, Biomedical Srl, Firenze, Italy) was used to localize the SILN (Figure 1).

After premedication and clipping of the surgical sites, dogs were positioned in dorsal recumbency and the surgical sites were aseptically prepared. Preoperative ultrasound was performed by a board-certified diagnostic imager, imaging resident or experienced ultrasonographer. As previously reported, the SILN were located using a linear ultrasound probe using a ventral approach (4–15 MHz frequency, MyLab™EightVET Esaote, Genova, Italy, and Samsung RS80 EVO, Samsung Heathcare, Kingsbury London, UK) [9]. When ultrasonographic detection of the SILN was difficult (i.e., small size and similar echogenicity relative to perinodal fat), the SILN was sampled by fine-needle aspiration and cytologically evaluated to confirm its nature. Under ultrasound guidance, the introducer needle was inserted through the skin and advanced into or adjacent to the LN. The optimal puncture site and the direction of the wire were previously agreed with the surgeon according to the surgical approach planned. In particular, the introducer needle was inserted as perpendicular as possible to the underlying LN and as close as possible to the intended site of the surgical incision. Once the tip of the needle was placed adjacent or in the middle of the LN, the wire was gently advanced by a second operator pushing the hook out of the needle. The hook was considered deployed when a black mark on the wire was visible through the Luer cone of the needle, and its anchorage to the tissue was checked by gentle traction. After deployment the hook cannot be re-inserted in the needle. The introducer needle was carefully removed over the wire, leaving the hook anchored to the target LN and the wire was visible on the skin surface (Figure 2) (Appendix A) [17].

The wire protruding from the operative field was trimmed approximately 2–5 cm from the skin insertion site and the area was covered with a sterile drape. The dogs were then transferred to the presurgical room for induction and then to the operating room with careful patient positioning to prevent dislodgment of the hook-wire. An alcohol-based solution was repeated over the surgical field prior to surgery. To avoid accidental dislodgment of the hook-wire, lymphadenectomy was performed before excision of the primary tumor or any other concomitant procedure.

A two to 4 cm skin incision was made in the inguinal region at entry site of the wire, avoiding any excessive manipulation or damage to the hook-wire. A combination of blunt and sharp dissection was used following the direction of the wire until the LN was located. Once the LN was isolated from the surrounding tissue, vessels of the lymph node hilus were ligated or cauterized to prevent hemorrhage. The LN was removed, and the surgical site was closed routinely.

Data were collected in a .xlsx (Microsoft excel for Mac, vers. 16.34) format file and analyzed with descriptive statistics. Continuous variables were presented as median and range, whereas categorical and ordinal variables were presented as absolute and relative frequency.

## 3. Results

Seventeen dogs for a total of 23 lymphadenectomies met the inclusion criteria. Breeds included crossbreed (five), Labrador retriever (three), pug (two), American Staffordshire (one), boxer (one), Cocker Spaniel (one), Entlebucher Mountain dog (one), golden retriever (one), Jack Russell terrier (one), Staffordshire bull terrier (one). There were nine male (five neutered) dogs and eight female (five neutered) dogs with a median age of 98 months (range 37–178). Body weight ranged from 6.8 to 56 kg (median 26.8). One (6%) dog had a BCS of four, three (18%) dogs had a BCS of five and six (35%) dogs had a BCS of six. Five (29%) and two (12%) dogs had a BCS of seven and eight, respectively. The most common primary tumor was cutaneous or subcutaneous mast cell tumor (15, 88%), followed by soft tissue sarcoma (2, 12%). These were most frequently located on hindlimbs (10, 59%), followed by trunk (3, 18%), prepuce (2, 12%), scrotum (1, 6%) and vulva (1, 6%). Superficial inguinal lymphadenectomy was performed in association with excision of the primary tumor in 14 (82%) cases. Nine cases (53%) had concurrent excision of other LNs; splenectomy was performed in two (12%) cases, and orchiectomy and thyroidectomy were performed in one (6%) case each. Median LN width was 3 mm (range 2–11). In fourteen (82%) dogs, the LN width was measured by ultrasound, in three (18%) by computed tomography. A single SILN was identified in the majority of dogs (16/17, 94%), and three SILN were identified on the same side in one dog. Median time for preoperative UGHW placement was 8 min (range 5–20) for each LN. Median lymphadenectomy time was 15 min (range 4–40). The hook-wire was placed perinodally in eighteen (78%) cases, intranodally in three (13%) cases, and out of the node in two (9%) cases. The UGHW allowed the localization of twenty-one (91%) of 23 SILN. In two (9%) cases, surgical dissection was started following the hook-wire, but a traditional lymphadenectomy was eventually performed. In particular, in one case, the hook was dislodged by wire traction during surgical dissection, whereas in another case, the hook-wire fragmented during surgical dissection and could not be retrieved despite the aid of intraoperative radiography. In these cases, surgical time (20 and 40 min, respectively) was longer compared to the rest of the population. SILN excision was successful in all the cases. During the follow up period (median 83 days, range 14–372), only one dog (6%) developed a self-limiting seroma at the lymphadenectomy site two days after surgery. This latter dog belonged to the twenty-one dogs that had a successful LN localization. No major intra- or post-operative complications were observed. The wire fragmentation that happened in dog n.17 resulted in no late complication 108 days after surgery. Histopathological evaluation of the SILN revealed no neoplastic cells in 10 (43%) LNs, whereas metastases were diagnosed in the remaining 13 (57%) LNs. Data are summarized in Table 1.

## 4. Discussion

Surgeons are more frequently asked to remove lymph nodes for diagnostic and therapeutic purposes, whether these are enlarged or normal size, even if these are not palpable and/or resulted non metastatic at the cytological examination [2,6]. In this study we demonstrated that the UGHW is a safe and effective localization method for the excision of non-palpable SILN.

As previously reported in healthy dogs, normal size SILN were considered palpable only in two (4%) of 50 cases, with a mean height and width of 3 mm and 6 mm [9], respectively, making LN identification and excision challenging. In the authors’ experience, surgical excision of non-palpable SILN is usually performed by a skin incision lateral to midline, approximately at the level of the cranial border of the pubis where the SILN is anatomically located; followed by a combination of blunt and sharp dissection to identify and excise the LN. To the best of the authors’ knowledge, information regarding successful excision rate, surgical time and intra-operative and short- and long-term post-operative complication of non-palpable SILN dissection is lacking in the veterinary literature. In the authors’ experience, identification and excision of non-palpable SILN may result in a time-consuming, extensive and unnecessary tissue dissection with an unpredictable successful rate of SILN excision. The UGHW localization method used in the present study allowed the excision of SILN in 91% of the cases with only two minor intra-operative and one short-term post-operative complications making the use of the UGHW for the excision of SILN a suitable method for a successful excision.

The preoperative ultrasound time, including visualization of the LN, cytological sampling (when indicated) and hook-wire placement observed in the present study was similar to the time of UGHW localization of impalpable breast nodules reported in women [18]. The time for UGHW placement may be similar to the time needed to excise a SILN without any localization method; however, in the authors’ opinion, UGHW placement allowed a high rate of successful SILN excision and reduced tissue dissection compared to a non-assisted lymphadenectomy. Further studies comparing assisted to unassisted SILN localization methods are warranted.

Only two intra-operative and one minor short-term post-operative complications occurred. The dislodgment of the hook in one case, and the fragmentation of the wire in the other did not affect LN excision which was successful in all dogs. Surgical time for SILN excision in these two cases (20 and 40 min) was longer than the median time for SILN excision. Hook-wire dislodgement and fragmentation are two of the most frequent complications associated with the procedure [12]. Recently in women, wire dislocation has been reported to occur in 27.5% of cases [18]; however, this was an infrequent event reported in our study. Planning with the surgeon the insertion point of the guide wire has reduced the wire dislocation rate from 45% to 10% in human patients [18]. As observed in our study, wire dislocation is infrequent if introduction site and direction of the wire is previously agreed with the surgeon. The frequency of retained wire fragments accounted for 0.2% in an 8-year retrospective study on 2498 breast nodule localizations in women [19]. In that study, patients had retained wire fragments for 1.5–11 years, and only in one case the fragment needed to be removed for patient discomfort. In our study, the dog with retained wire fragment was followed for more than four months before the manuscript was written and no complications were observed.

In people, infection rate has been reported to range from 0 to 1% in patients where the guide wire was placed the day before the surgery and in patients that were moved from the radiological room (where the wire-guided was placed) to the operating room [20,21]. Since infections associated with wire-guided placement seemed infrequent in people, we could speculate that also in dogs moved from ultrasonography to the operating room, the risk of infection is low. In this study population, no dogs experienced post-operative infection of the surgical site. To reduce contamination, the authors suggest covering the wire protruding from the skin with a sterile drape.

In twenty-one of 23 SILN, the hook-wire was placed either peri- or intranodally and the placement seemed not to influence node localization. In the two cases where dislodgement occurred, the hook-wire guided the surgeon to the target LN which was located immediately dorsal to the tip of the hook and the SILN was excised without extending the surgical dissection. Unfortunately a direct comparison with wire-guided lymphadenectomy in people is not possible, however, the successful rate of non-palpable breast and pulmonary lesion excision with the aid of preoperative wire-guided localization is higher than 95% in people, confirming that it is of great help for surgeons in the majority of cases [14,15,22,23].

Fifty-seven percent of the LN were diagnosed as metastatic, resulting in tumor stage migration from solitary tumor masses to locoregional disease. As previously reported, this highlights the superior accuracy of histology versus cytological evaluation and the important contribution of surgery to the tumor staging work-up [1,2,6]. The results of the present study suggest that the use of a minimally invasive SILN localization method represents an adequate method for lymphadenectomy of non-palpable SILN.

This study revealed that the UGHW localization method is associated with a high success rate for the excision of non-palpable SILN in dogs. The UGHW placement is an easy procedure, comparable to an ultrasound-guided fine-needle aspiration with the needs of an ultrasonographer and a second operator that will deploy the hook from the introducer needle.

The fact that LN excision using UGHW localization was performed by five surgeons with different levels of experience, suggests that this procedure may be easily replicated. We would also point out that planning with the surgeon the preferred location where the wire should enter the skin may result in a smaller skin incision, less extensive dissection, and faster node identification and excision.

The present study has some limitations: no control group was included in the study design, making any direct comparison with traditional LN excision not possible. Dogs were followed for a minimum of fourteen days; no long-term complications were evaluated, therefore, it is possible that late onset complications have been missed. The number of dogs included in this study was too small to statistically evaluate any relationship between variables.

## 5. Conclusions

In conclusion, the present study showed a high successful rate of SILN excision using a UGHW localization method. The UGHW localization method has shown to be a fast and easy method to perform. The UGHW placements and the SILN excisions were associated with infrequent and minor complications making the procedure suitable for further prospective and larger case-control studies to confirm our findings.

## Figures and Tables

**Figure 1 animals-10-02314-f001:**
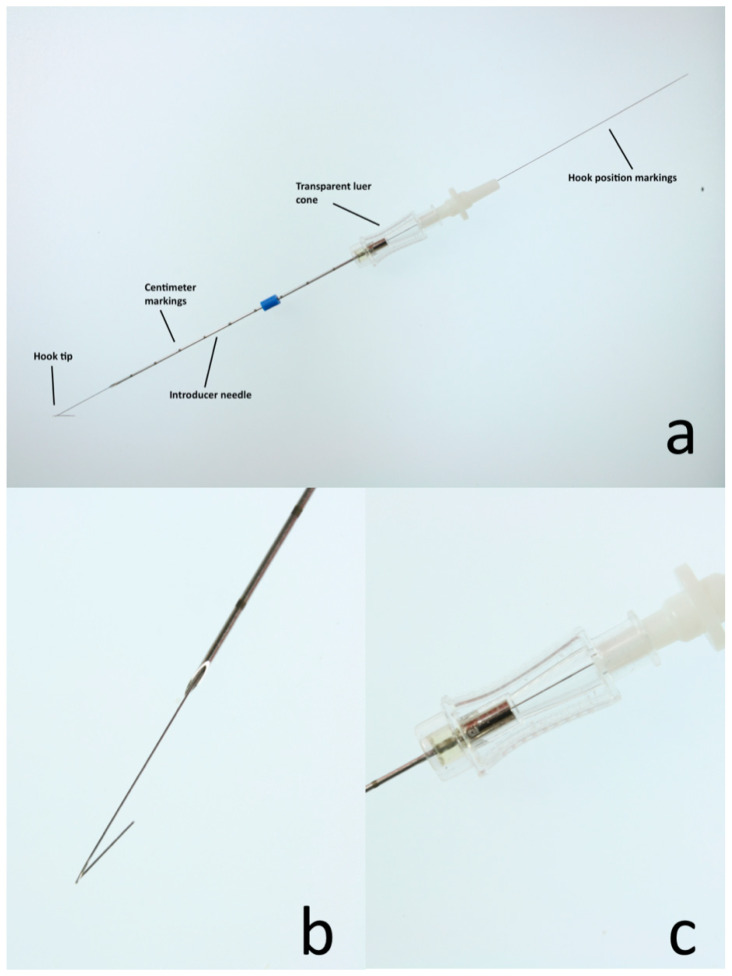
The figure shows a non-repositionable localization hook-wire needle used in the present study. (**a**): The hook tip securely keeps the device in place. The localization needle has centimeter markings for greater precision. The luer cone facilitates the wire insertion and allows the detection of the black mark on the wire to indicate the hook position. (**b**): Higher magnification of the hook tip. (**c**): Higher magnification of the luer cone.

**Figure 2 animals-10-02314-f002:**
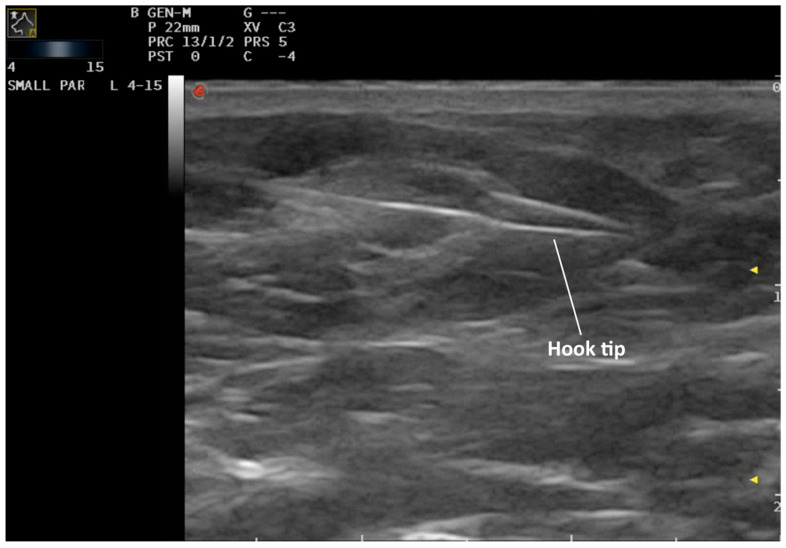
Ultrasound-guided hook-wire placement. From left to right: the figure shows the wire passing through the superficial inguinal lymph node and the hook tip close to the node.

**Table 1 animals-10-02314-t001:** Signalment and clinical data regarding ultrasound-guided hook-wire localization and lymph node excision of 17 dogs enrolled in the study.

Case	Breed	Age (mo)	Sex	BW (kg)	BCS (n/9)	Tumor Type and Location	SILN Side	SILN Width (mm)	Concomitant Surgery	UGHW Time (min)	Sx Time of LN Exc. (min)	Hook Location	Complications	LN Metastasis	FU (Days)
1	JRT	134	M	10.7	8	MCT; L thigh	L	3	M	5	8	I	No	N	191
							L	4		5	4	P		N	
							L	4		6	4	P		N	
2	Pug	96	FS	9.6	8	MCT; R distal hindlimb	R	2.5	M	5	10	P	No	N	50
3	Cross.	178	MN	46.7	7	STS; L distal forelimb	L	5 *	No	15	15	P	No	N	43
4	Labrador	125	F	37	7	mMCT; axilla, flank, sternum	R	4	M	20	40	O	Wire dislocation	N	24
5	Cross.	111	MN	31.7	7	MCT; R thigh	R	5 *	M	15	10	P	Seroma	Y	111
6	Pug	91	FS	8.6	5	MCT; R thigh	L	2.8	M	5	15	P	No	Y	42
							R	3.4		5	15	P		Y	
7	Cross.	84	MN	20.4	7	STS; L thigh	L	5	M	10	10	P	No	Y	45
							R	2.5		10	10	P		N	
8	Cross.	98	M	6.8	4	MCT; R hindlimb	R	3	M	10	25	P	No	Y	29
							L	2		10	15	P		Y	
9	SBT	121	MN	23.7	6	MCT; scrotal	L	2.7	No	5	25	P	No	N	135
							R	2.5		5	15	P		Y	
10	Boxer	68	FS	31	6	MCT; vulva	L	3 *	M	10	15	P	No	Y	83
11	EMD	37	M	56	6	MCT; R thigh	R	6	M	15	16	I	No	Y	363
12	Cross.	38	F	11.6	5	MCT; L thigh	L	4	M	6	5	P	No	N	372
13	Labr.	89	F	31.7	6	MCT; L flank	L	5	M + S	11	16	P	No	Y	359
14	Golden	126	FS	34.2	7	MCT; R tarsus	R	11	M + LN	8	7	I	No	Y	14
15	AST	118	M	37.3	6	mMCT; R Thigh, preputial	R	3	M + S	7	6	P	No	N	234
16	Cocker Sp.	108	MN	17.2	6	MCT; Preputial	L	2.2	M + S + O	10	15	P	No	Y	57
17	Labrador	71	FS	26.8	5	MCT; trunk (L chest)	L	3	T	6	20	O	Wire fragmentation	Y	108

BW: body weight; BCS: body condition score; SILN: superficial inguinal lymph node; LN: lymph node; UGHW: ultrasound-guided hook-wire; Sx time of LN exc.: surgical time of lymph node excision; FU: follow up; Specific acronymus per column: Breed: JRT: Jack Russell terrier; EMD: Entlebucher Mountain dog; AST: American Staffordshire terrier; Cocker Sp.: Cocker Spaniel; Cross.: crossbreed; SBT: Staffordshire bull terrier; Sex: M: male; MN: male, neutered; FS: female, spayed; F: female; Tumor type: MCT: mast cell tumor; mMCT: multiple mast cell tumor; STS: soft tissue sarcoma; Tumor location: L: left; R: right; SILN side: L: left; R: right; SILN width: * measured with computed tomography; Concomitant surgery (additional surgery to SILN excision): M: mass excision; S: splenectomy; O: orchiectomy; T: thyroidectomy; Hook location: I: intranodal; P: perinodal; O: out of the node; LN metastasis: Y: yes; N: no.

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
