# Peer review of "Ultrasound-Guided Hook-Wire Localization for Surgical Excision of Non-Palpable Superficial Inguinal Lymph Nodes in Dogs: A Pilot Study"

_animals, 2020, doi:10.3390/ani10122314_

Round 1
Reviewer 1 Report
I thank the authors for having considered the observations and for having integrated the elements necessary for a better understanding of the procedures adopted.
Reviewer 2 Report
Authors followed my my minor suggestions. The manuscript can be accepted in the present form.
This manuscript is a resubmission of an earlier submission. The following is a list of the peer review reports and author responses from that submission.
Round 1
Reviewer 1 Report
The paper describes the application of an ultrasound-guided hook-wire in order to facilitate the SILN excision in case of therapeutic or diagnostic lymphadenectomy. Despite the surgical technique looks interesting with possible further development, inclusion criteria and rationale for SILN excision needs to be better clarified.
Find below my comments:
Which was your rationale to excise the SILN? Did you get an anatomical criteria? Did you get a nodes map? Please clarify in the inclusion criteria and provide a comment in the discussion section.
Ln 68 – please describe the device and include a figure.
Ln 73 – inclusion criteria: please clarify the anatomical region you included
Ln 73- how did you get the diagnosis? Did you grade the tumours?
Ln 84 - please clarify what you mean with few millimeters. Give a range in case of perinodal hock
Ln 87-88 - please list the possible major complications
Ln 91 - any possible long-term complications?
Ln 94 – non-repositionable, you mean single use? Please clarify
Ln 100-101 – when indicated..please clarify indications for Fna
Ln 102-104 – please provide a figure of the ultrasonographic procedures
Ln 104 – please specify the recumbency during the needle insertion. The needle should inserted perpendicular to which axis? The video shows an oblique direction. Please clarify. Do you consider any further direction?
Ln 111- am quite concern about the patient’s double transfer from ultrasonography to the theatre in terms of potential infection and/or wire migration. Please provide a comment in the discussion section.
Ln 117- what do you mean with small skin incision? In which anatomical area did you perform the surgical approach? I suppose it should be approximately the same. Please specify.
Ln 136-137- which other nodes? Why?
Ln 150- 152- Dogs had a mean 83 days follow up. how many re-check during the follow up period?? Specify in the main doc that dog with seroma was within the 21/23 group.
Ln 153-154 No metastases in 43% of cases. Can you exclude any further nodes involvement?
Ln 206-208 …As observed in our study, wire dislocation would be infrequent if the optimal puncture site and the direction of the wire is previously agreed with the surgeon according to the surgical approach planned. Please clarify: which surgical approach?
Table 1 – in some cases tumors were far from the inguinal nodes. As on the top I would like to understand your rationale for SILN excision. IT would be interesting getting the tumor grading. Please provide.
Thank you for your research
Author Response
Response to Reviewer 1 Comments
Reviewer’s comments in black
Authors’ responses in red
Which was your rationale to excise the SILN? Did you get an anatomical criteria? Did you get a nodes map? Please clarify in the inclusion criteria and provide a comment in the discussion section.
Dear reviewer, the study did not have the intent of identify the sentinel LN, although the authors are fully aware of its importance and differences when compared to the regional LN. The rational for excising the SILN was tumour anatomic location and normal anatomical lymph node drainage of the peritumoural tissue. Overall, the SILN were excised as they were the regional lymph nodes. Despite this is may be a limitation in accuracy of tumour staging, the scope of the study was to describe and discuss a surgical technique, not to focus on sentinel lymph node and lymph node mapping.
Ln 68 – please describe the device and include a figure.
Dear reviewer, we added figure 1.
Ln 73 – inclusion criteria: please clarify the anatomical region you included
Dear reviewer, we think that the anatomical region has been already stated in the materials and methods section (line 73-74: “Dogs affected by different tumor histotypes presented at two referral hospitals for diagnostic or therapeutic lymphadenectomy of SILN with the aid of UGHW placement between August 2019 and May 2020 were enrolled”). However, we edited the text to reduce misinterpretation. (line 78)
Ln 73- how did you get the diagnosis? Did you grade the tumours?
Dear reviewer, cases come from oncology orientated practice, one of them is also an accredited training center of the ECVIM-CA oncology, so we can guarantee that diagnosis and staging work-up were performed according to guidelines. However, since the aim of this manuscript was to describe a method for localization-assisted lymphadenectomy, we believe that data regarding modality of diagnosis and tumor grading are not essential information and that these will instead contribute in diluting the study content rather than adding relevant info. We have therefore elected not to include these data; however, these can be made available to the reviewer upon request.
Ln 84 - please clarify what you mean with few millimeters. Give a range in case of perinodal hock
We edited the text as suggested (line 89)
Ln 87-88 - please list the possible major complications
We edited the text as suggested (line 92-93)
Ln 91 - any possible long-term complications?
Dear reviewer, we described a pilot study focused on the clinical application of a localization technique for superficial inguinal lymph node excision and short- and medium-term complications. To evaluate long-term complications, the study should be re-designed standardizing clinical re-evaluation of dogs. We add a comment in the limitation section (line 548-549)
Ln 94 – non-repositionable, you mean single use? Please clarify
Non-repositionable means that once the hooked wire is advanced out from the needle and deployed into the tissue, it is not possible to re-insert it in the needle or modify its position. After deployment the hook is anchored to the tissue and can be removed only with surgical excision. A comment was added in the text (line 133-134)
Ln 100-101 – when indicated..please clarify indications for Fna
We edited the text as suggested (line 119-120)
Ln 102-104 – please provide a figure of the ultrasonographic procedures
Dear reviewer, we added figure 2.
Ln 104 – please specify the recumbency during the needle insertion. The needle should be inserted perpendicular to which axis? The video shows an oblique direction. Please clarify. Do you consider any further direction?
Dear reviewer, the recumbency was specified at line 99 “…were positioned in dorsal recumbency and…”. Regarding the direction of the needle insertion, it should be “…as perpendicular as possible to the underlying LN...” as stated in line 109-110. No other directions have been taken into consideration.
Ln 111- am quite concern about the patient’s double transfer from ultrasonography to the theatre in terms of potential infection and/or wire migration. Please provide a comment in the discussion section.
Thank you for this comment. The protocol we designed has taken into consideration these possible issues. The patients were anesthetized, clipped and aseptically prepared. A sterile drape was placed over the surgical area and the patient was transferred to the ultrasound room. The preoperative ultrasound was performed aseptically (sterile gloves, sterile cover for the ultrasound probe, alcohol was used rather than gel). After deployment of the hook-wire the area was covered with a sterile drape and the patient was moved to the operating room. At this point a second alcohol-based preparation was used on the surgical area carefully avoiding any dislodgment of the hook-wire. We agree with the reviewer’s concern however we did not experience any increased complications using this protocol. We added a comment in the discussion section (280-286)
Ln 117- what do you mean with small skin incision? In which anatomical area did you perform the surgical approach? I suppose it should be approximately the same. Please specify.
We edited the text as suggested (line 147)
Ln 136-137- which other nodes? Why?
Dear reviewer, other lymphadenectomies were part of the staging or treatment of the single case. We would not further specify these procedures because they would not add any essential information to this study
Ln 150- 152- Dogs had a mean 83 days follow up. how many re-check during the follow up period?? Specify in the main doc that dog with seroma was within the 21/23 group.
We edited the text at line 197-198
Ln 153-154 No metastases in 43% of cases. Can you exclude any further nodes involvement?
Dear reviewer, we cannot exclude any further metastases, however, since it was not an aim of this manuscript the authors retained this information not essential for the reader.
Ln 206-208 …As observed in our study, wire dislocation would be infrequent if the optimal puncture site and the direction of the wire is previously agreed with the surgeon according to the surgical approach planned. Please clarify: which surgical approach?
We edited the text as suggested.
Table 1 – in some cases tumors were far from the inguinal nodes. As on the top I would like to understand your rationale for SILN excision. IT would be interesting getting the tumor grading. Please provide.
Dear reviewer, as stated in the introduction “The aim of the study was to describe the clinical application of an ultrasound-guided hook-wire (UGHW) LN localization method for the excision of non-palpable SILN in dogs and to report successful excision rate and surgical complications associated with this technique”. The rationale for SILN excision was not reviewed and completely demanded to the oncologist in charge case by case. As for comment regarding line 73, tumor grading and rational for SILN excision would not add any essential information to the scope of this study

Reviewer 2 Report
Pierini and colleagues describe a study on ultrasound-guided hook-wire localization for surgical excision of non-palpable superficial inguinal lymph nodes in dogs. I feel that this is a nice and short paper that highlights the benefit of a not so frequently used procedure in Veterinary medicine. I have only minor comments which are highlighted below.
General comments:
- I see that there is material online, perhaps Authors could consider to add a picture/scheme to explain to the reader the methodology used.
Specific comments:
- Line 36: change into “no majour…” and “or’ and you can remove one of the “major”
- Line 45: perhaps “clinical” decision making?
- Line 81: the BS result of the animals to be moved into results.
- Line 121: war routine would mean routinely performed? Please change
- Line 133 and following for the tumour types: due to the small numbers perhaps better to express: N/totalN rather than % with decimals
Author Response
Reviewer’s comments in black
Author’s responses in red
General comments:
- I see that there is material online, perhaps Authors could consider to add a picture/scheme to explain to the reader the methodology used.
- Dear reviewer, we added figure 1.
Specific comments:
- Line 36: change into “no majour…” and “or’ and you can remove one of the “major”
We edited the text as suggested
- Line 45: perhaps “clinical” decision making?
We edited the text as suggested
- Line 81: the BS result of the animals to be moved into results.
Dear reviewer, if the comment regards “BCS, ranged from 1 to 9”, the range was not a result but a method of scoring. Results regarding BCS has already been reported in the result section (line 139-141). We edited the text to avoid misinterpretation
- Line 121: war routine would mean routinely performed? Please change
We edited the text as suggested
- Line 133 and following for the tumour types: due to the small numbers perhaps better to express: N/totalN rather than % with decimals
If we have understood the reviewer refers to “. The most common primary tumor was cutaneous or subcutaneous mast cell tumor (15, 88%), followed by soft tissue sarcoma (2, 12%). These were most frequently located on hindlimbs (10, 59%), followed by trunk (3, 18%), prepuce (2, 12%), scrotum (1, 6%) and vulva (1, 6%).”, in this case the numbers refer to (N, %), and there are no decimals in this sentence. As stated in the materials and methods section “categorical and ordinal variables were presented as absolute and relative frequency”; when decimal have been reported, units and decimals are separated by point and not by comma. We would leave to the editor a final decision about the presentation of these findings.
